# The Physics of Entropic Pulling: A Novel Model for the Hsp70 Motor Mechanism

**DOI:** 10.3390/ijms20092334

**Published:** 2019-05-11

**Authors:** Rui Sousa, Eileen M. Lafer

**Affiliations:** Department of Biochemistry and Structural Biology, University of Texas Health Science Center, San Antonio, TX 78229-3900, USA; lafer@uthscsa.edu

**Keywords:** chaperones, Hsp70, Hsp110, entropic pulling, power-stroke, brownian ratchet, DISAGGREGATION, motor proteins, J cochaperones, protein translocation

## Abstract

Hsp70s use ATP to generate forces that disassemble protein complexes and aggregates, and that translocate proteins into organelles. Entropic pulling has been proposed as a novel mechanism, distinct from the more familiar power-stroke and Brownian ratchet models, for how Hsp70s generate these forces. Experimental evidence supports entropic pulling, but this model may not be well understood among scientists studying these systems. In this review we address persistent misconceptions regarding the dynamics of proteins in solution that contribute to this lack of understanding, and we clarify the basic physics of entropic pulling with some simple analogies. We hope that increased understanding of the entropic pulling mechanism will inform future efforts to characterize how Hsp70s function as motors, and how they coordinate with their regulatory cochaperones in mechanochemical cycles that transduce the energy of ATP hydrolysis into physical changes in their protein substrates.

## 1. Introduction

Chaperone proteins were so dubbed because they “prevent inappropriate interactions” [1], a designation that aptly captures what was initially believed to be their primary or exclusive function: to prevent non-specific aggregation, but not to disrupt aggregates. However, as understanding of chaperones grew, it was discovered that these proteins could resolubilize protein aggregates [2,3,4], and that an Hsp70 chaperone, together with specific assisting cochaperones in carefully titrated amounts, could even solubilize amyloid fibrils, whose stability had been expected to preclude resolubilization [5]. The ability of particular Hsp70 chaperone–cochaperone systems to break up aggregates was added to their abilities to disassemble defined protein complexes (e.g., clathrin coats [6]) or translocate proteins into the ER [7] or mitochondria [8] as examples of the ability of these systems to act as motors that transduce the chemical energy of ATP to move or physically alter their substrates. Efforts to understand the mechanism of these motors, particularly in the context of protein translocation [9,10], focused initially on the two models—Brownian ratchet or power-stroke—that have long delimited thought on how molecular motors act. However, it also led to the proposal of a model, dubbed *entropic pulling* [11], that represents a genuinely novel break in the debate between the two more familiar models. Subsequent experimental evidence has supported entropic pulling [12], but it’s unclear if this model is well understood in the chaperone community. In this review we describe the three models that have been used to explain Hsp70 motor mechanisms, and expand on the description of the entropic pulling model, using simple examples to clarify the basic physics. Evidence in favor of this model, and inconsistent with the other competing other models, is also discussed. Ideally, this review will enhance understanding and inform new experimental approaches to characterizing how Hsp70s work and coordinate with cochaperones.

## 2. Models for Hsp70 Motor Mechanism

### 2.1. The Usual Suspects: Power-Stroke or Brownian Ratchet

NTP burning molecular motors have usually been described as working by one of two mechanisms: a power-stroke, or what is variously described as the Brownian (or thermal) ratchet or conformational (or state) selection mechanism [13]. In the power-stroke model, one of the steps in an NTP hydrolysis cycle (binding, hydrolysis, or dissociation of the NDP or phosphate) results in a conformational change in a protein that directly and mechanically moves the overall reaction in a specific direction. For example, Hsp70s are motor proteins that can pull apart protein complexes or aggregates, or pull proteins through translocation pores. The power-stroke mechanism for these proteins is easiest to understand in the context of the latter where we can imagine that ER or mitochondrial specific Hsp70s can bind to an unfolded segment of protein that is emerging through a translocation pore on the lumenal or matrix side of the ER or mitochondrion, and then undergo a conformational change, coincident with a step in the ATP hydrolysis cycle, that directly pulls the protein further into the organelle (Figure 1A).

Power-stroke models are appealing because the molecular motors seem to act like miniaturized versions of the macroscopic machines with which we are familiar. However, the microscopic world differs from the macroscopic one in that molecules are in constant movement, driven by unceasing molecular collisions and fluctuations. A mechanism to generate motion is therefore not required, but a mechanism to drive this motion in a particular direction is. From such considerations emerge the Brownian/thermal ratchet or conformational/state selection mechanisms. Again using protein translocation as our example, consider that an unfolded protein passing through a translocation pore can randomly slide back and forth in the pore, driven by collisions and fluctuations. If Hsp70s that can bind to sequences in the unfolded protein are present on one side of the translocation pore, they will bind those sequences as they emerge and block backwards sliding of the protein from that point. Repeated cycles of such random sliding coupled to binding by Hsp70swill eventually move the entire protein into the organelle (Figure 1B).

### 2.2. Entropic Pulling: A Novel Model for Motor Protein Mechanisms

Discussions of motor protein mechanisms have usually been circumscribed by these two models. It was therefore refreshing when Goloubinoff and De Los Rios proposed an entirely novel mechanism for the Hsp70 motor that they termed *entropic pulling* [11,14]. The entropic pulling model notes that Hsp70s bound to ATP are recruited to their substrates by cochaperones called J proteins which usually load the Hsp70s onto flexible peptide segments emerging from a larger macromolecular structure such as a translocation pore, a protein aggregate, or protein complex like a clathrin coat. Upon ATP hydrolysis, the Hsp70 releases its interaction with the J cochaperone and clamps down on the peptide segment. In this configuration, the Hsp70’s freedom of movement and entropy is restricted by the immediately adjacent macromolecular structure. If it were to move away from this wall, pulling the bound peptide segment along as it does so, it would gain freedom of movement and entropy (Figure 1C). This favorable entropy change generates a large directional force that Goloubinoff and De Los Rios proposed to be the mechanism by which Hsp70s can translocate proteins or disrupt protein aggregates and complexes.

#### Trees vs. Forests, or Molecular-Kinetic vs. Thermodynamic Descriptions of Entropic Pulling

But how, mechanically, does entropic pulling actually generate a directional force large enough to unfold a protein or pull a polypeptide out of an amyloid fibril? It can be easier to understand this by referencing a simpler, analogous system. Consider how we are introduced to the phenomenon of gas pressure, which is explained as being generated by collisions of gas molecules with the walls of the vessel in which they are contained (Figure 2A). This is the molecular-kinetic (trees) description. We could also give a thermodynamic or forest description by noting that expansion of the gas is a molecular process governed by ΔG = ΔH−TΔS. Since (ideally) gas expansion is purely entropy driven, we can reduce this to ΔG = −TΔS. But ΔG is also work, which is also force multiplied by the distance over which the force acts, so we then have: ΔG = −TΔS = work = −F*ΔX, or F = TΔS/ΔX. We can therefore also describe gas pressure as arising because, upon expansion, the gas will gain motional freedom and entropy, and will therefore generate a force equal to the first derivative of the free energy (entropy) change with respect to the distance increment (Figure 2B).

Entropic pulling represents a thermodynamic or forest description for how Hsp70s generate force. Thermodynamic descriptions can be more rigorous than molecular-kinetic ones, and they also allow us to estimate the potential forces generated. By calculating the greater motional freedom and entropy gained by a tethered Hsp70 as it moves incrementally away from close apposition to a wall, Goloubinoff and De Los Rios estimate that a single Hsp70 can generate a force of 20 pN (Figure 2C). But, in reprising our analogy of the different descriptions of gas pressure mechanisms, we would note that most students probably understand and retain the lesson that *“gas pressure is due to collisions between gas molecules and vessel walls”* better than *“gas pressure arises because, upon expansion, gas molecules would gain motional freedom and entropy, which results in a force being generated that is proportional to the first derivative of the entropy increase with respect to the distance increment”*. Molecular-kinetic descriptions facilitate understanding of how, at the molecular level, forces are generated. Since most biochemists and molecular biologists are deeply invested in reductionist, atomic level descriptions of biological processes, it can be important to provide such tree-level molecular descriptions alongside the thermodynamic forest. We can therefore note that an Hsp70 tethered to a flexible peptide emerging from a protein wall will be jostling around and colliding with that wall. This will impart a momentum (an entropic push) against the wall and an equal and opposite momentum against the Hsp70 that will then exert an entropic pull on the peptide segment to which it is tethered. As these entropic pushes and pulls force the Hsp70 and wall apart, the frequency of collisions between the wall and the chaperone will diminish, as will the forces (Figure 2C).

### 2.3. The Hsp70 Mechanochemical Cycle

Recent studies of the Hsp70 force generation mechanism, using Hsp70-mediated disassembly of clathrin coats as a tractable system, supported the entropic pulling model, and resulted in the proposal for the general Hsp70 mechanochemical cycle shown in Figure 3 [12]. In keeping with the use of analogies to enhance understanding, we can describe this cycle as working very much like a compressed gas cooling system. There is an ATP-burning enthalpic compression phase whose function is to load Hsp70s into constrained spaces where the entropic pulling/pushing forces exerted by collisions and repulsion between the flexibly tethered Hsp70s and constraining walls are greatest, and there is an entropically driven expansion phase in which these forces drive the Hsp70s and constraining walls apart. 

#### 2.3.1. Mechanisms for J Cochaperone Loading of Hsp70 in Constrained Spaces

To be efficient this cycle requires that the J cochaperone load Hsp70 onto its protein substrates close to the constraining wall(s), and that the nucleotide exchange factor (NEF), which unloads Hsp70 from its substrates, does so when the Hsp70 is farther from these walls, towards the end of the expansion phase of the cycle. The mechanisms by which this geometric specificity is achieved are not fully understood, but some answers may be in view. Notably, J cochaperones display both substrate binding domains and Hsp70 binding J domains, and must interact simultaneously with the protein substrate and Hsp70 to load the latter onto the former. This may provide a mechanism to drive Hsp70 loading into constrained spaces.

For example, auxilin, the J cochaperone which recruits Hsp70 to clathrin coated vesicles, binds to the clathrin N-terminal domain but loads Hsp70 onto an Hsp70 binding site at the clathrin C-terminus [15,16,17]. In an individual clathrin triskelion (the unit from which semi-spherical clathrin coats are assembled) the N-terminal auxilin binding site and the C-terminal Hsp70 binding site are far apart, and auxilin is not efficient at loading Hsp70 onto isolated triskelia (Figure 4A). However, in a clathrin coat, the N-terminal auxilin-binding domains of one triskelion are close to the C-termini of their neighbors. Auxilin loading of Hsp70 onto clathrin coats is efficient and presumably exploits this proximity to load Hsp70 *in trans* into the constrained space under the vertex of a proximal triskelion where the C-terminal Hsp70 binding segments are located (Figure 4A). As another example, protein translocation pores are large complexes that incorporate proteins with J domains into their architectures [7,8]. This provides a mechanism for loading Hsp70 close to the complex and onto the translocating protein just as it emerges from the pore (Figure 4B). Finally, for reactions like protein aggregate disruption, which utilize more generalist J cochaperones that have more promiscuous substrate binding properties, we imagine that such aggregates display multiple, extended hydrophobic loops or termini on their surfaces that can be bound by either J cochaperones or Hsp70. A J cochaperone binding to such an aggregate would likely load an Hsp70 onto nearby sites, again resulting in a situation in which the Hsp70 is tethered on a peptide segment immediately adjacent to a motionally constraining wall (Figure 4C).

#### 2.3.2. Mechanisms for Biasing NEF Unloading of Hsp70s from Accessible Spaces

We would expect that protein binding in the constrained spaces into which Hsp70s are recruited during the compression phase of the mechanochemical cycle (Figure 3) will be sterically less favorable than binding in more accessible spaces. Indeed, when antibody binding to a FLAG tag introduced into the clathrin C-terminus was measured, it was found that moving the tag away from the clathrin coat walls by inserting 10 or 25 AA upstream of the tag site progressively increased binding rates by more than 2- and 3-fold, respectively (Figure 5) [12]. The J cochaperone requirement to interact simultaneously with protein substrate and Hsp70 during Hsp70 loading provides a mechanism to overcome this bias for binding to more accessible sites, rather than spatially constrained ones. Hsp70 NEFs, however, do not need to bind simultaneously to protein substrates and Hsp70 when they act on the latter, and therefore lack such a mechanism. This should bias them to acting upon Hsp70s that are at the end of the expansion phase of the mechanochemical cycle (Figure 3), when they are more accessible. Such biasing may be enhanced for the NEFs most effective at supporting disaggregation—the Hsp110s—which are usually larger than J cochaperones and display long disordered loops or termini that add disproportionately to their hydrodynamic radii [19] and would inhibit close approach of these proteins to constrained spaces. Gao et al., for example, observed that the DNAJB1 J cochaperone bound closely to the surface of synuclein fibrils, allowing detection of a FRET signal between the synuclein and DNA JB1, which covered the fibril surface densely with particles 2–3 nm in size. In contrast, the APG-2 (Human Hsp110) NEF bound much more sparsely as particles 5–10 nm in size that were at greater distance from the fibril surface [5]. Differences in hydrodynamic radii and in how J cochaperones and NEFs interact with Hsp70s may, in whole or in part, provide the mechanisms that bias J cochaperone loading of Hsp70s into constrained spaces during the compression phase of the mechanochemical cycle, and NEF unloading of Hsp70s from more accessible spaces towards the end of the expansion phase.

#### 2.3.3. The Energy of ATP is Harnessed at the Loading and Unloading Steps of the Cycle

When analogizing a macroscopic task, like busting up a rock pile, to a microscopic task, like breaking up a protein complex or aggregate, it’s compelling to assume that the hard, energy consuming steps will be the same. For the rock pile, this is not the initial step of setting one’s feet next to the pile, nor is it unsetting one’s feet and moving closer to the pile after our efforts have broken up a part of it (Figure 6, step 1 and 3). Instead, we would be burning most of the ATP in vigorous whacking with our pick, and it is the analogous step in the microscopic process that often consumes most of our mechanistic interest just as it is expected to consume most of the ATP (Figure 6, step 2). However, the microscopic world is very different. To begin with, positioning a protein near a protein complex or aggregate and keeping it there is not easy. As anyone doing single molecule studies knows, the only way to keep a protein on a given spot is to stick it there. If the protein is free in solution, collisions and fluctuations will rapidly move the protein away. Necessarily, if fixing the protein to a spot next to the substrate is hard, in that it requires significant stabilizing interactions, then releasing if from that spot to allow it to continue its work will also be hard and energy consuming. On the other hand, the step analogous to whacking with the pick need not consume ATP at all. It is enough if the initial step, in which the Hsp70 is placed close to the substrate, is done by tethering the Hsp70 to a flexible peptide emerging from the substrate. Fluctuations and collisions will constantly jostle the tethered Hsp70, allowing it to act as a wrecking ball against the substrate to which it is tethered (Figure 6, step 2).

#### 2.3.4. ATP Binding Stabilizes an Otherwise Unstable Hsp70 Conformation that Disrupts Interactions with NEFs and Substrates, and Forms the Binding Site for the J Cochaperone

Consideration of the Hsp70 cycle indicates that it is, in fact, at the steps of loading and unloading that ATP energy is used. The NEF-mediated unloading step, for example, must disrupt the Hsp70*ADP*substrate complex. This is a stable, long-lived complex, but the most effective Hsp70 NEFs (the Hsp110 cochaperones) are able to disrupt it by making an extensive set of interactions with the Hsp70 nucleotide binding domain (NBD), forcing the latter to open and release bound ADP [20,21]. This does not yet unload Hsp70 from its protein substrate, but merely swaps one stable interaction (Hsp70*ADP) for another (Hsp70*NEF). However, with the nucleotide binding site now empty, ATP can bind and induce closure of the NBD and displacement of both the interactions with NEF and the protein substrate (Figure 7). In the absence of a NEF, ADP can still spontaneously (if slowly) dissociate from Hsp70, and ATP can then bind and drive the conformational change that results in substrate release. In the absence of any nucleotide, Hsp70 assumes a conformation similar to that seen when ADP is bound, indicating that this is the more stable, lower-energy conformation, and that the ATP induced conformation is the higher energy state [22]. This indicates that *the energy of ATP binding is used to stabilize an otherwise unfavorable conformational state that disrupts interactions between Hsp70s and NEFs, and between Hsp70s and their protein substrates*. 

Crystal structures of Hsp70s complexed with ATP indicate that the ability of ATP, but not ADP, to stabilize this conformation is a consequence of the exceptionally high density of interactions between the protein and γ-phosphate which encompass fully half of the 14 H-bonds made between the Hsp70 and ATP [23]. The Hsp70*ATP conformation not only disrupts interactions with substrates and NEFs, it also forms the binding site for the J cochaperone and is more compact than the ADP or nucleotide-free conformation (Figure 7) [24]. Assumption of this conformation is essential for appropriate J mediated recruitment of the Hsp70, and its more compact nature may facilitate recruitment of the Hsp70 into constrained spaces. We may therefore also conclude that *the energy of ATP binding is used not only to unload Hsp70 from its NEFs and substrates, but also to enable the interaction with the J cochaperone that results in Hsp70 loading*. 

### 2.4. The CRITICAL functions of the Cochaperones are Geometrically Specific and Timely Loading and Unloading of Hsp70s

In the mechanochemical cycle proposed in Figure 3, the primary functions of the cochaperones are loading and unloading of Hsp70 from substrates. However, in addition to domains that interact with Hsp70s, all J cochaperones [26], and the Hsp110 NEFs, display domains that interact directly with protein substrates [19]. Different classes of J cochaperones have been found to be differentially effective in supporting Hsp70 mediated soluble protein or amyloid disaggregation, and the Hsp110 NEFs have been found to be more effective at supporting disaggregation than NEFs that lack protein substrate binding domains [3,4,5,27,28,29,30]. Since all of these J cochaperones carry out Hsp70 loading, while all the NEFs can execute unloading, the observation that they nevertheless differ in their ability to support Hsp70 mediated disaggregation suggested that the critical functions of these cochaperones extend beyond Hsp70 loading and unloading. In particular, it was proposed that Hsp110 NEFs might cooperate with Hsp70s as part of disaggregation engines that could, for example, “walk” along and disaggregate proteins by coordinately binding and releasing substrate interactions with the Hsp110 or Hsp70 substrate binding domains (SBDs) [2]. 

However, these models have generally not been supported by subsequent studies [30]. For example, persistent interactions between Hsp110 and Hsp70s in the presence of ATP and protein aggregates, which would be consistent with a progressive disaggregation engine, have not been observed. Instead, characterization of amyloid fibril disaggregation detected abundant J cochaperone association with long fibrils as well as with shortened fibrils that represent intermediates in the disaggregation process. Substantial amounts of Hsp70 were also detected in association with these species, but Hsp110 was present at much lower levels that required mass spectroscopy for detection. Moreover, when Hsp110 was added to fibrils that had been pre-loaded with Hsp70, there was no detected accumulation of Hsp70*Hsp110 complexes on the fibrils but, instead, immediate and immeasurably fast unloading of the Hsp70s [5]. 

Detection of this complete and rapid unloading led to a proposal that the unloading step itself represents a power-stroke in which a transient chaperone complex of Hsp110, Hsp70, and the J cochaperone generates a powerful pull on the fibrillar components coincident (or immediately preceding) the dissociation of all these proteins from the fibril [5]. However, J cochaperones bind to Hsp70*ATP, while NEFs bind to Hsp70*ADP, so a complex formed by direct interactions between all three proteins seems unlikely. Moreover, Hsp110 and Hsp70 bind with 1:1 stoichiometry [20,21], which, in any mechanism that depends on a power-stroke delivered by a complex between these two proteins, would imply that disaggregation should be maximized when Hsp110 and Hsp70 are near a 1:1 ratio. But this is not what is observed: Hsp110 supported Hsp70-mediated disaggregation is maximal at substoichiometric ratios of Hsp110 to Hsp70, with ratios of 1:10 or 1:20 being optimal for fibrillar disaggregation [4,5]. The reduced effectiveness of higher Hsp110 levels in supporting disaggregation may, however, be readily understood if we consider that optimal levels of this NEF are those that allow for efficient J cochaperone loading of Hsp70s, and that bias unloading of Hsp70s to occur towards the end of the expansion phase of the Hsp70 mechanochemical cycle. Conversely, high NEF levels can lead to excessive unloading that excessively reduces the number, lifetime and proximity of Hsp70s bound to substrates. This can reduce the entropic pulling/pushing force that is exerted on the substrate and slow disaggregation. 

#### 2.4.1. Optimal Amounts of Different NEFs in Disaggregation Reactions Correlate with their Unloading Activity

In support of this, NEFs like Bag1 are less efficient than Hsp110s at *both* supporting fibrillar disaggregation *and* at unloading Hsp70s from fibrils, and they exhibit a higher optimal concentration in disaggregation assays. Thus 0.1 μM Hsp110 is *more* effective than 0.2 or 1 μM Bag1 in supporting disaggregation, but 1 μM Hsp110 is *less* effective than 0.1 μM Hsp110 or 1 μM Bag1, which is, in turn, *more* effective than either 0.2 or 5 μM Bag1 [5]. Bag1 does not have an SBD, which indicates that Hsp70 unloading alone, in the absence of interactions between the NEF and protein substrate, is sufficient to support disaggregation. The correlation between unloading activity and optimal concentration for disaggregation argues that Hsp70 unloading is the critical function for NEF supported disaggregation, and that unloading must be optimized so that there is enough to efficiently unload Hsp70s from stalled positions at the end of the expansion phase of the mechanochemical cycle but not so much that Hsp70s are unloaded from forcefully productive positions at the start of this phase.

#### 2.4.2. Faster J Cochaperone Loading of Hsp70 Correlates with More Effective Disaggregation

Similarly, the observation that fibrillar disaggregation is supported by the DNAJB1, but not the DNAJA1, cochaperone could mean that additional activities, beyond simply loading of Hsp70s, are important for the functions of these cochaperones in disaggregation. However, DNAJB1 loads Hsp70 onto fibrils more rapidly than DNAJA1 [5], so it may simply be kinetic differences that account for their differences in supporting disaggregation. We could also imagine that different J cochaperones might load Hsp70 in relatively different positions on the fibril, with the J cochaperone that loads the Hsp70 in the most constrained space (closest to the body of the fibril) being most effective at disaggregation.

#### 2.4.3. Interactions with the SBDs of J Cochaperones and NEFs Could Assist Disaggregation by Shielding Proteins from Reaggregation

While these considerations argue for Hsp70 loading and unloading as being the critical activities through which J cochaperones and NEFs support Hsp70 mediated disassembly of protein aggregates and complexes, they do not rule out that the SBDs of these cochaperones might contribute secondarily to disaggregation by, for example, transiently binding to hydrophobic regions of polypeptides that become exposed during disaggregation so as to block their reaggregation. In this light, the properties of the Hsp110s are of note. Hsp110s are Hsp70 homologs with ATP-binding NBDs and SBDs that bind extended hydrophobic protein sequences much like canonical Hsp70s do. Their affinities for these sequences are modulated by ATP and ADP much as Hsp70s are [31]. However, Hsp110s do not undergo the large ATP hydrolysis dependent conformational changes that cause the helical subdomain of the SBD to clamp down on bound proteins. Instead, the Hsp110 SBD appears to be in a constitutively open state, with its helical subdomain displaced onto a site on the NBD [32]. Perhaps due to this inability to clamp down and trap substrates, the rates of protein substrate dissociation from Hsp110 are much faster than from Hsp70 [31], and Hsp110s alone cannot unfold proteins or disassemble aggregates, but can act as holdases that shield unfolded or misfolded proteins from aggregating [33]. The ability of Hsp70 to unfold proteins, or disassemble aggregates and complexes, may therefore be related to its ability to clamp down and form long-lived complexes with its protein substrates so that, following ATP hydrolysis and release of the J cochaperone interaction, the Hsp70 is free to act like a tethered wrecking ball.

#### 2.4.4. Hsp70 Oligomerization or Persistent Associations with Cochaperones could Increase Entropic Pulling Forces

Hsp70 cochaperones could also enhance entropic pulling/pushing forces by binding to and conferring added bulk to the Hsp70 wrecking ball. In support of this, antibody binding to Hsp70 has been shown to accelerate Hsp70 mediated clathrin coat disassembly [12]. More physiologically, Figure 3 shows a side-pathway (step1b) in which an Hsp70 is loaded onto a substrate, and releases but stays near the J cochaperone. This may occur, for example, if the Hsp70 is working on a recalcitrant substrate, like a very stable aggregate. In this case the persistent proximity of the Hsp70 to the J cochaperone allows the latter to load a second Hsp70 onto the first, exploiting the fact that the exposed linker connecting the Hsp70 NBD and SBD in the Hsp70*ADP state (Figure 7A) has features like a canonical Hsp70 substrate. Since each added Hsp70 presents a new interdomain linker that can be bound by another Hsp70, this J cochaperone and ATP dependent oligomerization of the Hsp70s may continue indefinitely [34] until either the Hsp70 at the end of the chain is too far from the J cochaperone, or until the substrate gives way to the greater entropic forces provided by increased numbers of Hsp70s and they begin to move away from the J cochaperone. The binding of either additional Hsp70 molecules or cochaperones can therefore augment the force generated by a single Hsp70. For J cochaperones and NEFs the scope of this effect may be limited, since J cochaperones don’t bind Hsp70 during the latter’s ADP and substrate bound force-generating phase of its cycle (Figure 3), and because NEFs interact only transiently with Hsp70 during this phase. However, the Hsp70 cochaperone called Hip forms a persistent interaction with and stabilizes Hsp70*ADP or Hsp70*ADP*substrate complexes [35,36,37]. Hip is therefore a good candidate for a cochaperone that can stably bind and add to the bulk of a substrate bound Hsp70, potentially enhancing its force generation ability. Consistent with this, we have found that Hip association with Hsp70s bound to clathrin coats accelerates coat disassembly (unpublished observations).

### 2.5. Entropic Pulling: Objections and Misconceptions

The ideas behind the entropic pulling/pushing or collision pressure model are entangled with a number of persistent misconceptions among biologists and biochemists that can make it hard to understand or accept. Here we address those that we have most typically encountered:

#### 2.5.1. This Model Assumes That Forces Are Generated by Collisions. But in the Crowded, Viscous Environment of the Cytoplasm Proteins Are Moving so Slowly, with Such Short Mean Free Path Steps, That They Cannot be Generating Collision Forces in the Same Way as Molecules Moving Rapidly in the Gas Phase

In fact, elementary physics informs us that (at constant temperature) molecules in a liquid phase, however viscous and crowded, collide with the same velocity and kinetic energy as they would in the gas phase. The latter statement is often greeted so skeptically that it can be necessary to call in authority to buttress it: “*It can be shown from the kinetic theory that the velocities of molecules of the dissolved substances are not affected by the molecules’ being in solution, but are equal to the velocities that they would have if they were in a gaseous state. Therefore, both the number and the intensity of the impacts of the molecules of the dissolved substances against the membrane are equal to the number and intensity of the impacts that one expects for a gas. The pressures exerted in both cases are therefore equal*” (from Enrico Fermi’s 1937 text on thermodynamics explaining the molecular-kinetic origin of osmotic force).

#### 2.5.2. A Mechanism Based on Local Dissipation of Entropy Cannot Deliver as Big a Force as a Power-Stroke

This objection seems related to the tendency to analogize macroscopic and microscopic worlds, which leads to the expectation that ATP driven microscopic power-stroke mechanisms analogous to the power-strokes delivered by big machines will necessarily be stronger than mechanisms based on molecular collisions and changes in local entropy. However, power-stroke motors like kinesin and myosin generate forces in the 4–9 pN range [38,39], while an Hsp70 is calculated to generate a maximal force of about 20 pN [11,14]. The force generated by kinesin or myosin molecules is exerted over a 6–8 nm step size [40,41], while the force generated by an Hsp70 falls rapidly as it moves 2–3 nm away from a restraining wall. The work done by these systems per ATP hydrolyzed is therefore similar, but the force mechanisms are appropriate for each: Hsp70s generate larger forces over shorter distances, appropriate for extracting a protein from an aggregate or unfolding a stable protein as it moves through a translocation pore, while kinesin or myosin exert a weaker force over a longer distance, appropriate for moving cargo through a cell. 

Since the 20 pN figure is an in silico calculation, it can be viewed skeptically, but experimental solution data in support of such a value emerges from measuring the osmotic forces generated by concentrated protein solutions. Such forces are the macroscopic product of microscopic collisions between the protein and the semi-permeable membrane, and they drive osmotic flows of solvent. This description of osmotic forces contradicts descriptions found in textbooks stating that flow of water through a semi-permeable membrane from a low to a high solute compartment occurs because more of the membrane’s surface is occluded on the high solute side, and therefore occludes water diffusion from that compartment to a greater degree than from the low solute compartment. Dr. Eric Kramer has waged a battle against this misconception (Osmosis Confusion: 60 Years and Counting, Scientific American 2013; Five Popular Misconceptions about Osmosis, American J. of Phys, 2012; Osmosis is not driven by water dilution, Trends in Plant Sci., 2013) [42,43,44], but the misconception persists. In fact, such flows, which are much faster than can be accounted for by diffusion [43], occur because solute molecules collide with and are repelled from the semi-permeable membrane. Collisions between solute and water molecules then transfer the momentum from the solute to the solvent, driving a flow of solvent away from the membrane and into the higher solute compartment. This is analogous to the collisions that a tethered Hsp70 makes with a constraining wall, resulting in a momentum on the Hsp70 away from the wall that is then transferred to the tether as a pull when the tether reaches maximum extension. 

Experimentally, it’s found that a 448 mg/mL solution of bovine serum albumin can generate over 5 atm of osmotic pressure, or about 0.5 pN per nm^2^ [45]. Since this pressure is present at every point in the solution, each albumin molecule in contact with the walls of the vessel is generating a force of about 30 pN against those walls. Local protein concentrations of 400–600 mg/ml are effectively obtained by tethering Hsp70 close to a wall. The in silico calculation of the microscopic entropic pulling force that an Hsp70 can generate is therefore consistent with a macroscopic analog—the osmotic forces generated by concentrated protein solutions—and shows that these entropic forces can exceed those generated by power-stroke mechanisms.

These osmotic forces are highly non-ideal: the 5 atm generated by a 448 mg/ml albumin solution is 20-fold greater than would be the case if albumin molecules could be treated as points obeying the ideal gas law. Much of this non-ideality is due to the volume of the albumin molecules and can be estimated from the Carnahan–Starling hard-sphere approximation [46], which, at the specified concentration, would predict a 5-fold greater force than calculated from the ideal case. The additional factor of 4 may be due to charge. At concentrations of about 450 mg/ml, the osmotic force is 1.25 atm at pH 4.5 (net charge on albumin = +4.5), 2.1 atm at pH 5.45 (net charge = −9.6), and 5.1 atm at pH 7.42 (net charge = −20.5) [45]. The observation that repulsive charge interactions increase osmotic force has potential implications for the Hsp70 mechanism. Hsp70s typically have pI values in the 5–6 range. At cytoplasmic pH they would usually carry significant negative charge that would be even greater in the more basic compartments (e.g., mitochondrial matrix) in which some Hsp70s function. Many of the substrates on which Hsp70s work (e.g., synuclein, clathrin) are also negatively charged, as are the membrane proximal spaces that Hsp70s approach during protein translocation. In addition, many Hsp70 cochaperones, especially the Hsp110s and Hip, exhibit unstructured loops and termini that carry a large excess of negative charge and numerous phosphorylation sites. Repulsive charge interactions between Hsp70s and substrates, between Hsp70s and membranes, between Hsp70s themselves, and between the cochaperones and all these species could therefore enhance the pulling/pushing forces that these systems generate.

The relationship between entropic pulling and osmotic forces reminds us of another mechanistic model—Peter Mitchell’s chemiosmotic hypothesis for ATP synthesis—that was also initially received with skepticism or lack of understanding. Just as the search for high energy chemical intermediates in the models competing with chemiosmosis was to prove fruitless, so has the search for disaggregation engines composed of Hsp70:cochaperone complexes. Instead, the force driving ATP synthesis was ultimately understood to be entropic and to originate from the dissipation of a proton concentration gradient. Similarly, entropic pulling can be described as a mechanism in which a high local protein concentration is achieved by tethering Hsp70s within a constrained space. As this high local concentration dissipates, pulling/pushing forces are generated. The development of these models shows how important fundamental thermodynamics and classical molecular-kinetic descriptions are for the understanding of all biological processes. 

#### 2.5.3. High Intracellular Protein Concentrations Will Nullify the Entropic Forces Generated by the Locally High Protein Concentrations Effectively Obtained by Tethering Hsp70s in Constrained Spaces

This objection needs to be answered semi-quantitatively. Intracellular soluble protein concentrations have been estimated to be approximately 250 mg/ml [47]. An average MW of 60 kD implies a protein concentration of about 4 mM, or an average distance between protein molecules of about 7.5 nm. However, the effective local concentrations generated by attaching Hsp70 molecules to short tethers can be much higher, corresponding to distances of 1–3 nm between molecules. This means that, while high intracellular protein concentrations can reduce the net increase in local protein concentration generated by Hsp70 tethering, and will therefore reduce the net force, the effective local concentration generated by Hsp70 loading is still substantially higher than in its surroundings and strong forces will still be generated.

### 2.6. Entropic Pulling Is a Novel Model and Elicits Novel Questions 

Power-stroke motors like myosin and kinesin utilize precisely coordinated interactions to move along defined macromolecular tracks (actin fibers in the case of myosin; microtubules for kinesin). This necessarily circumscribes their functions, since they cannot actively move except on these specific tracks. In contrast, what the entropic pulling mechanism for Hsp70 lacks in precisely coordinated moving parts that might, for example, pass substrates between different chaperones and cochaperones, it gains in general applicability. A more precise and coordinated mechanism might be difficult to adapt to the array of reactions—from resolubilization of heterogeneous aggregates and amyloid fibrils, to disassembly of symmetric complexes and translocation of proteins into organelles—that Hsp70s execute. On the other hand, a system based on attaching a dynamic wrecking ball to a peptide emerging from a substrate seems capable of generating force in a variety of contexts. 

This novel model also changes the focus of our experiments, from questions about the machine-like coordination of chaperone–cochaperone complexes, to questions about the timing and geometric specificity of Hsp70 loading and unloading in the mechanochemical cycle. For example, we cite data above showing that an antibody binds more rapidly to sites that have been moved further away from a wall of clathrin protein (Figure 5). Would NEFs similarly unload Hsp70s that are more accessible because they are dangling on longer tethers, and could this mechanism bias NEFs to unload Hsp70s towards the end of the expansion phase of the mechanochemical cycle? Do different NEFs (e.g., Hsp110s vs. Bags) exhibit different degrees of this bias due to their size or charge, and can this, as well as intrinsic differences in unloading kinetics, explain some of the differences in how these NEFs support disaggregation? Do the large, charged unstructured loops and termini displayed by some NEFs add volume and charge to these molecules to increase this bias? Are differences in the ability of different J cochaperones to support disaggregation due to differences in Hsp70 loading kinetics and/or a bias for some J cochaperones to load Hsp70 closer to constraining walls? Do optimal J and NEF cochaperone types and amounts in a disaggregation reaction correspond to conditions that maximize the total entropic pulling/pushing pressure being generated by Hsp70s as might be detected, for example, as a maximum in the total FRET signal between labeled Hsp70s and the aggregate? Are repulsive charge interactions between 70s, constraining walls and cochaperones important in generating force, and could phosphorylation and cellular pH contribute to this? 

## 3. Summary and Conclusions

Entropic pulling represents a genuinely novel model for motor protein action that goes beyond the classical Brownian ratchet or power-stroke mechanisms. The challenges to understanding and accepting this model include persistent misconceptions that proteins in the crowded cytoplasmic environment move too slowly to generate significant collision forces, and that entropic forces associated with dissipation of local concentration gradients will necessarily be smaller than those generated by power-strokes. The former misconception is at odds with fundamental physics, and the latter is contradicted by in silico calculations and measures of osmotic forces generated by concentrated protein solutions. Deploying both molecular-kinetic and thermodynamic descriptions of this model can aid in understanding. A J cochaperone disengages from an Hsp70 after loading it onto a substrate, leaving the Hsp70 as a tethered wrecking ball that can generate collision forces that translocate proteins or break up protein complexes or aggregates. Together with NEFs, Hsp70s and J cochaperones create a mechanochemical cycle that encompasses an ATP-burning compression phase that loads Hsp70s onto substrates in constrained spaces, and an entropically driven expansion phase that drives physical changes in those substrates. This cycle will work most effectively when J cochaperones rapidly load Hsp70s into the most forcefully productive, constrained spaces, while NEFs primarily and rapidly unload them from more accessible spaces reached at the end of the cycle’s expansion phase. This can explain the observed cochaperone concentration optima for in vitro disaggregation reactions, and implies that understanding how Hsp70s and cochaperones cooperate in these reactions will depend more on understanding how the cochaperones achieve timely and geometrically specific Hsp70 loading and unloading, rather than on the details or precise coordination of cochaperone:substrate or cochaperone:Hsp70 interactions. 

## Figures and Tables

**Figure 1 ijms-20-02334-f001:**
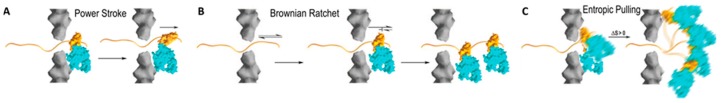
Different models for Hsp70 motor action in the context of protein translocation through an import channel. (**A)** Power-Stroke. Hsp70 (in cyan and yellow) binds the translocating protein (yellow) and a site on the import channel (grey). Coincident with a step in its ATP hydrolysis cycle it undergoes a conformational change that generates a directed pull on the protein. (**B**) Brownian ratchet. An unfolded polypeptide can slide back and forth in the channel but Hsp70 binding on one side blocks backwards sliding. Repeated cycles of such sliding and Hsp70 binding can eventually move the entire protein through the channel. (**C)** Entropic pulling. Loading of Hsp70 onto the translocating protein close to the import channel restricts Hsp70 freedom of motion. Movement of the Hsp70 away from the channel increases its motional freedom and entropy, generating an entropic pull on the translocating protein. Adapted from ref. [13].

**Figure 2 ijms-20-02334-f002:**
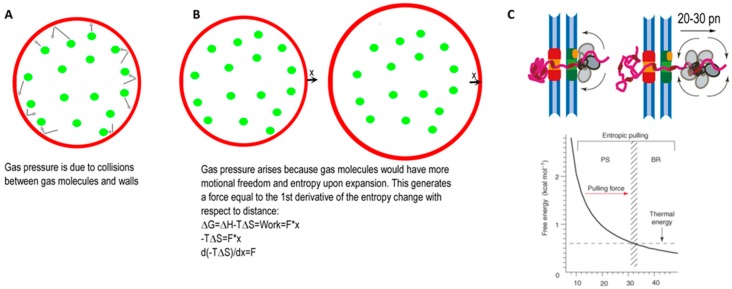
Molecular-kinetic vs. thermodynamic descriptions of gas pressure illuminate the entropic pulling mechanism. (**A**) The molecular-kinetic description of gas pressure ascribes it to collisions between vessel walls and gas molecules. (**B**) The thermodynamic description ascribes it to the entropy increase the gas molecules gain upon expansion. (**C**) Entropic pulling represents a thermodynamic description for how Hsp70s generate force. The force can be estimated as suggested by panel B: by calculating the entropy of the Hsp70 when it is held close to the wall by its peptide tether vs. when it is further from the wall. The decrease in free energy and force follow the curve below the cartoon. Alternatively, we could give a molecular-kinetic description of entropic pulling by pointing out that, when tethered close to the wall, the Hsp70 will frequently collide with and impart a push against the wall, resulting in an equal and opposite push on the Hsp70 which will then pull on the tether. As the Hsp70 moves away from the wall, the frequency of these collisions and the forces will decrease. Panel C adapted from ref [14].

**Figure 3 ijms-20-02334-f003:**
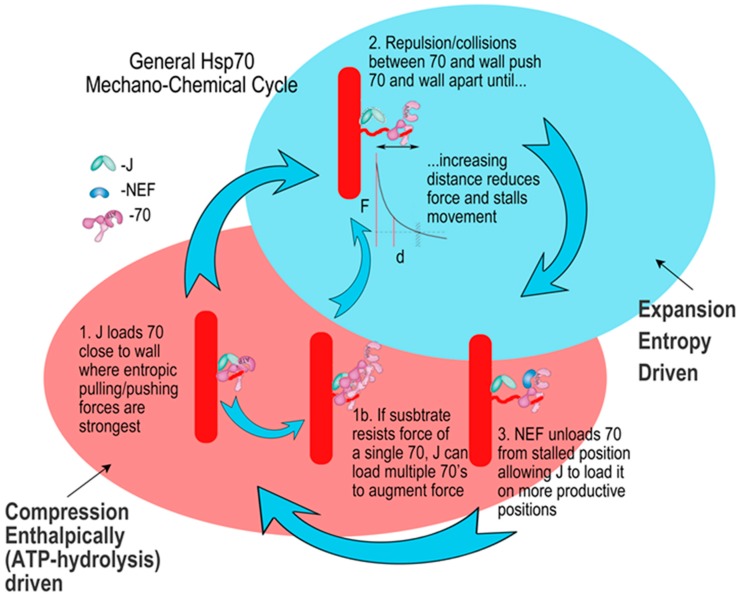
The general Hsp70 mechanochemical cycle can be analogized to a compressed gas cooling system. In the energy consuming compression phase, a J cochaperone loads Hsp70*ATP onto a peptide tether adjacent to constraining protein walls and the ATP is hydrolyzed. In the entropy driven expansion phase the J cochaperone releases the Hsp70*ADP, and collisions/repulsive interactions drive Hsp70 and the walls apart. Forces diminish as the distance between Hsp70 and the walls grows, until a nucleotide exchange factor (NEF) induces ADP dissociation so that ATP can bind and drive release of Hsp70 from the peptide tether, resetting the cycle. Adapted from ref. [12].

**Figure 4 ijms-20-02334-f004:**
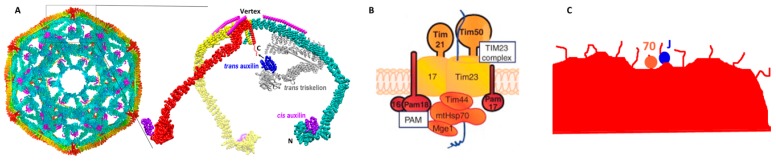
Mechanisms of J cochaperone loading of Hsp70s into constrained spaces. (**A**) Loading Hsp70 onto clathrin cages. Structure of a clathrin:auxilin coat with auxilin in magenta, the interior of the coat in cyan and the exterior in yellow and orange, and with the vertices that mark the C-terminal region of clathrin in red. The expanded section shows a triskelion from this coat with the 3 clathrin heavy chains in cyan, yellow, and red, and the clathrin light chains in magenta (pdb 1XI5 [17]). Part of a neighboring triskelion is shown in grey. Auxilin J domains (purple and blue) bind the globular N-terminal domain of clathrin (“N”) and load Hsp70 onto the flexible C-terminal tails (“C”) located in the constrained space under the vertex of a clathrin triskelion. Loading of Hsp70 onto isolated triskelia is inefficient, possibly because of the distance between the clathrin N- and C-termini. But in a clathrin coat, triskelia are positioned with their N-terminal domains close to the C-termini of their neighbors, allowing for efficient auxilin mediated loading in trans. (**B**) Loading of mtHsp70 onto translocating proteins. Architecture of the mitochondrial translocation pore. Tim44 and Pam18 recruit Hsp70 to the mouth of the pore on the matrix side of mitochondrion and position it to bind translocating proteins as they emerge [8]. (**C**) Loading of Hsp70 onto protein aggregates. A protein aggregate displays exposed hydrophobic loops and termini that will have binding sites for J cochaperones and Hsp70s, resulting in Hsp70 loading close to the body of the aggregate. In all three cases ATP hydrolysis causes the Hsp70s to clamp down on their substrate polypeptides and disengage the J cochaperones (auxilin, Pam18), while the Tim44:70 interaction is disrupted by the interaction with the substrate [18]. The result is Hsp70 tethered next to constraining wall(s) with no other interactions holding it in that space.

**Figure 5 ijms-20-02334-f005:**
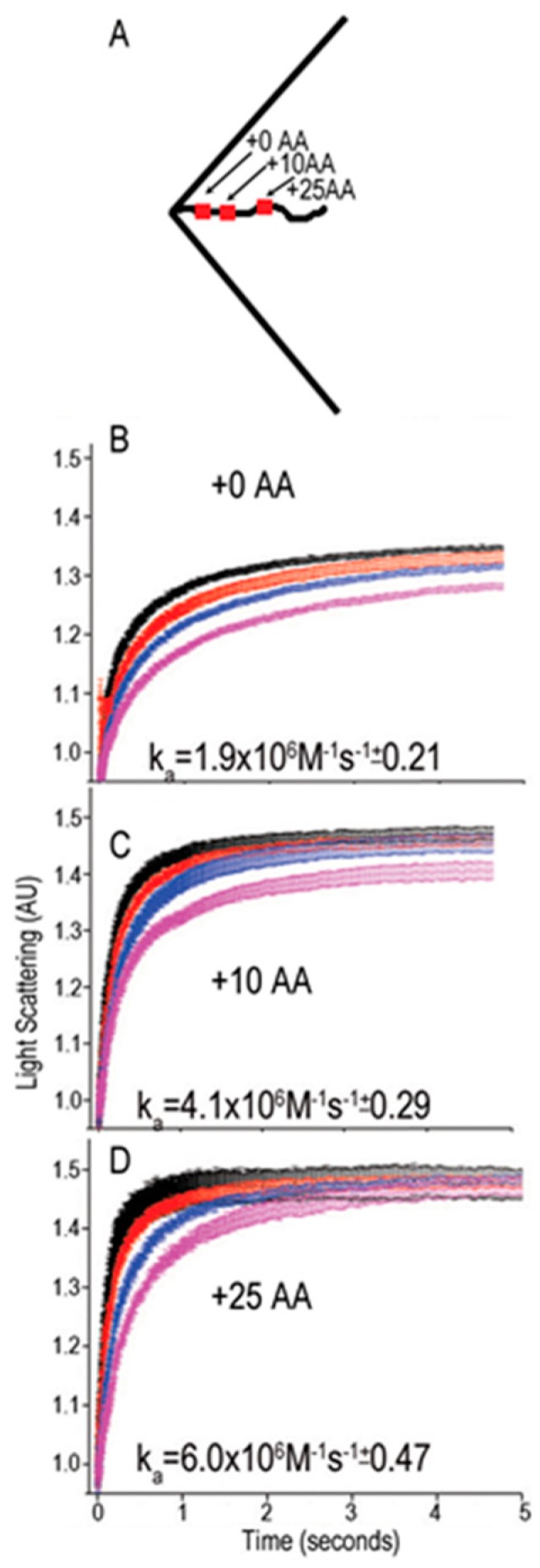
Accessibility could bias NEFs to bind Hsp70s towards the end of the expansion phase of the mechanochemical cycle. (**A**) Clathrin triskelia in which the Hsp70 binding site was replaced with a FLAG tag were constructed and used to assemble clathrin coats. Triskelia in which this tag was moved away from the constrained space under the triskelion vertex were also constructed by inserting 10 or 25 amino acids between the tag and upstream sequences. (**B**,**C**) The binding of anti-FLAG Fabs at 0.25 (magenta), 0.5 (blue), 1.0 (red), and 2.0 (black) μM to clathrin coats assembled with FLAG-tag triskelia in which the tag has been moved downstream by 0 (panel B). Ten (panel C) or 25 (panel D) AA were measured by stop-flow light scattering. Increasing the tag’s accessibility by moving it away from the constrained space under the vertex progressively increased the association rate (k_a_). Panels B–C adapted from ref. [12].

**Figure 6 ijms-20-02334-f006:**
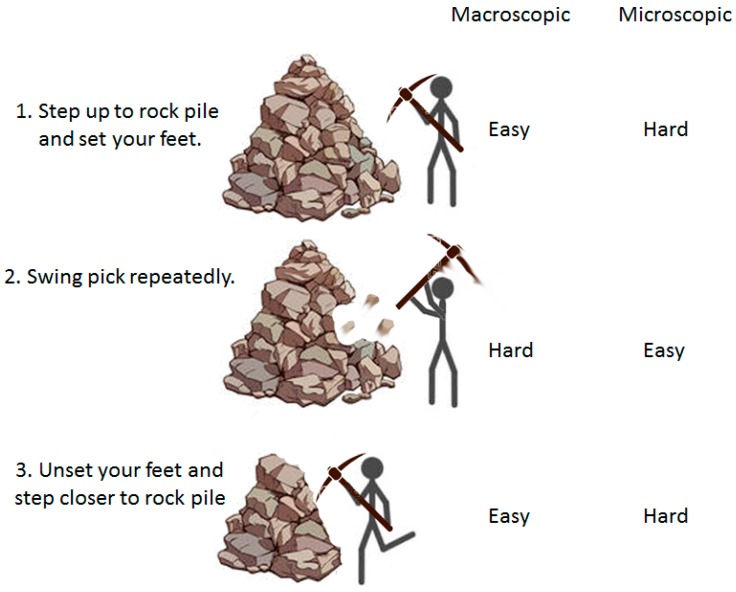
Different steps in breaking up an aggregate are differentially difficult at the macroscopic and microscopic levels. At the macroscopic level stepping up to the aggregate and setting one’s feet (step 1) is easy, as is unsetting one’s feet and moving closer to the pile (step 3) once our efforts have broken up part of it, but the intermediate step of whacking at the pile is energy consuming and hard. At the microscopic level the reverse is true: fixing a protein near an aggregate (step 1) is hard and requires stabilizing interactions, which necessarily means that step 3, in which these interactions are disrupted, will also be hard and energy consuming. Step 2, however, is easy and need not consume ATP since collisions and fluctuations provide the motive forces that drive the pick against the aggregate.

**Figure 7 ijms-20-02334-f007:**
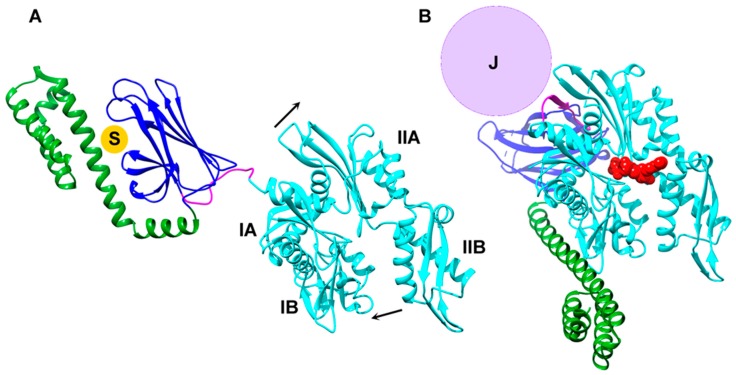
ATP energy is used to stabilize an Hsp70 conformation that disrupts interactions with NEFs and protein substrates, and forms the binding site for the J cochaperone. (**A**) Structure of bacterial Hsp70 (DnaK; 1KHO.pdb [25]) in the ADP state with the nucleotide binding domain (NBD) in cyan, the interdomain linker in magenta, and the β-sheet and helical subdomains of the substrate binding domain (SBD) shown in blue and green, respectively. This is the intrinsically more stable conformation which is also assumed in the absence of nucleotide [22], and which traps substrate proteins (“S”) as extended peptides between the helical and β-sheet parts of the SBD. (**B**) Structure of Hsp70 in the ATP conformation (1BQ9.pbd [23]). ATP binding induces closure of the NBD by hinging of lobe IIB towards lobe IB, and lobe IIA away from IA, as indicated by arrows in panel A. Closure of the NBD disrupts interactions with NEFs, which bind and stabilize open conformations of the NBD. Widening of the space between lobes IA and IIA allows the interdomain linker to bind in this space and extend the lobe IIA β-sheet. These changes create binding sites for the J cochaperone (which binds in the space defined by sphere labeled “J”) and for the β-sheet and helical parts of the SBD, resulting in opening of the latter and release of bound substrates.

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
