# Peer review of "The Physics of Entropic Pulling: A Novel Model for the Hsp70 Motor Mechanism"

_ijms, 2019, doi:10.3390/ijms20092334_

Round 1

Reviewer 1 Report

The authors submitted a manuscript on “The Physics of Entropic Pulling: A Novel Model for the Hsp70 Motor Mechanism”. 

The topic of this paper is of scientific importance and close to  the scope of “Molecular Sciences”.

The abstract introduces the topic, but it is not clear showing the need of the review. 

In the present contribution, the entropic pulling model for the generation of proteins dissembling forces by Hsp70 was described.  The introduction section provides an overview of the topic and introduces the objective of the review, but it is not so clear.

The whole paper seems not so well organized with a not so clear scope to be achieved.

Throughout the text, some inaccuracies can be found:

-        The text layout should be justified and spacing should be checked (example lines 46-82)

-        All the figures are blurry and their resolution should be improved

-        The numbers of references are repeated twice    

Author Response

Response to reviewer comments:

Formatting issues:

1. Text justification & spacing: We have corrected this (left & right justification; uniform 1.5x line spcing) wherever we could identify problems referred to by the reviewer.

2. Figures blurry:  The set of figures currently embedded in the ms is lower res than final set so as to facilitate the uploading &  transfer of the manuscript for review.  Higher resolution figure can be sent for final publication.

3. Reference numbers are repeated twice:  We've been unable to avoid this when using endnote to format references with the electronic word template provided for us to write our review in.  If provided with guidance on what endnote formatting option will eliminate this problem we will do so.  Otherwise, it can probably be taken care of by the journal upon publication.  We would note that any other formatting issues may also be dealt with this way.

More substantively, the reviewer critiqued both the language and style in the ms, and its overall organization. As none of the other 3 reviewers had similar criticisms, and as this reviewer was not specific in describing the issues with language, style and manuscript organization, we had some difficulty in understanding the nature of the required changes.  However, we have gone through the manuscript as carefully as possible and made changes to meet the reviewer's critique when we could see how to do so...

We thank the reviewer for his time and efforts in reading our manuscript.

Reviewer 2 Report

The article is devoted to the interpretation of the entirely novel model of the mechanism for the Hsp70 motor vehicle proposed by De Los Rios et al. (2006), which was termed entropic pulling. Prior to this, there were two models, namely power-stroke and Brownian ratchet models, but none of them explained all the features of the Hsp70. Entropic pulling model unifies the two Hsp70 functions of protein disaggregation and mitochondrial protein import.

This review is very relevant because, as the authors note, although entropic pulling model is supported by experimental studies, it is not well understood among scientists studying these systems. It is of great importance because it is important to understand that the laws of thermodynamics lie not only in the basis of physical processes, but also in the basis of all processes occurring in living matter, starting with the formation of hydrogen bonds by water molecules and ending with the formation of non-membrane organelles.

In the works of De Los Rios et al., as well as in the early works of the authors, all functions of Hsp70 (mediating the correct folding of de novo synthesized proteins, the translocation of proteins across membranes, the disassembly of some native protein oligomers, and the active unfolding and disassembly of stress-induced protein aggregates) are considered in terms of the laws of thermodynamics. In this paper, the authors not only provide an overview of articles relating to this problem, but also try to convey to the reader the significance of such ideas.

I have only one small remark: the authors need to emphasize that the laws of thermodynamics are universal, that they can and should be used to explain the work of other chaperones, proteins, and in general the organization of cellular space. Otherwise, one would think that the laws of thermodynamics relate only to the functioning of the Hsp70.

Author Response

To accommodate the reviewer's comment that we emphasize that thermodynamic and molecular-kinetic considerations should be used in understanding all biological processes, not just Hsp70 function we have added the following paragraph to the manuscript (near the end of section 2.5):

The relationship between entropic pulling and osmotic forces reminds us of another mechanistic model--Peter Mitchell's chemiosmotic hypothesis for ATP synthesis--that was also initially received with skeptisism or lack of understanding.  Just as the search for high energy chemical intermediates in the models competing with chemiosmosis was to prove fruitless, so has the search for disaggregation engines composed of Hsp70:cochaperone complexes.  Instead, the force driving ATP synthesis was ultimately understood to be entropic and to originate from the dissipation of a proton concentration gradient.  Similarly, entropic pulling can be described as a mechanism in which a high local protein concentration is achieved by tethering Hsp70s within a constrained space.  As this high local concentration dissipates, pulling/pushing forces are generated.  The development of these models shows how important fundamental thermodynamics and classical molecular-kinetic descriptions are for the understanding of all biological processes. 

We thank the reviewer for his efforts in reading our manuscript.

Reviewer 3 Report

Authors did a good job reviewing Hsp70 motor mechanism. They have described and discussed the three different models that have been used to explain Hsp70 motor mechanism in a very instructive way. Thus, remarking the entropic pulling model which is the latest theory to explain Hsp70 mechanism. I accept the revision in its present form. 

Author Response

Response to reviewer queries:

We thank the reviewer for his time and efforts.  As this reviewer requested no revisions we have no further response to the review. 

Reviewer 4 Report

The paper is reviewing the different models for Hsp70 motor motility. The entropic pulling mechanism is a very interesting new way to understand mechanical movements and it's also intriguing at the same time. 

I think the authors need to illustrate a couple of points:

1- Looking at motors such as dynein and kinesin, the power stroke model is the one which is widely accepted now as a model to understand how these motors move over microtubules. The authors need to comment on why this would be a different model then Hsp70 motility for example? Is there any evidence in the literature that dynein or kinesin also adopts the entropic pulling model?

2- The analogy between the entropic pulling model and busting up a rock pile to understand how energy is harnessed is a bit confusing. Figure 6 is suffering the same problem, I literally failed to see any differences between both sides of the figure except the word easy and hard which doesn't require really the figure to mention. Unless the figure is showing something different, it's really unnecessary. Therefore, I recommend rewriting this part in a clearer way and also to remove the figure or use a different figure to compare the two situations. 

Author Response

Response to reviewer queries:

To accommodate the reviewer's request for more discussion of kinesin/myosin vs. Hsp70 mechanisms we have added the following bolded text to the review: 

Power-stroke motors like myosin and kinesin utilize precisely coordinated interactions to move along defined macromolecular tracks (actin fibers in the case of myosin; microtubules for kinesin).  This necessarily circumscribes their functions, since they cannot actively move except on these specific tracks.  In contrast, what the entropic pulling mechanism for Hsp70 lacks in precisely coordinated moving parts that might, for example, pass substrates between different chaperones and cochaperones, it gains in its general applicability.  A more precise and coordinated mechanism might be difficult to adapt to the array of reactions--from resolubilization of heterogeneous aggregates and amyloid fibrils, to disassembly of symmetric complexes and translocation of proteins into organelles--that these systems work on.  On the other hand, a system based on attaching a dynamic wrecking ball to a peptide emerging from a substrate seems capable of generating force in a variety of contexts. 

    2. To address the reviewer's confusion regarding figure 6 we have modified the figure so that it no longer contains a duplication of almost identical images.  

    3.  We have re-run spell check & corrected any typos/grammar etc. we could find...

We thank the reviewer for his time and efforts in reading our manuscript.

Round 2

Reviewer 1 Report

I thank the Authors for their effort to improve the manuscript (accomplished) and for their response.

However, I would like to reply to this sentence "the reviewer critiqued both the language and style in the ms, and its overall organization. As none of the other 3 reviewers had similar criticisms", suggesting that the reason could be found on  what marked by other reviewers, namely "I don't feel qualified to judge about the English language and style".

Moreover, what stated "this reviewer was not specific in describing the issues with language, style and manuscript organization" is not true, as regarding the manuscript organization it was clearly referred to the scope of the manuscript and regarding the English language it is not reviewer's work to underline/change any sentence or typo found in the text.

I hope this clarify better...

Author Response

We thank the reviewer again for his or her time and efforts.  We did not mean to negatively characterize the reviewer's comments and apologize if our response was so interpreted.

If we interpret the reviewer's comment below correctly, the reviewer wishes to clarify that his comments regarding manuscript organization was directed to (the reviewer's word) the 'scope' of our manuscript and (as indicated by the # star rankings) its current significance to the field.

Such a critique can be difficult to respond to, as it may reflect subjective evaluations of the state of the field, the community's understanding, etc...  We based our decisions regarding the scope and significance of our ms on the following considerations:

When we received an e-mail requesting a submission for this special chaperone issue, it was explicitly stated that what was wanted was a paper that addressed the ideas laid out in our 2016 NSMB paper.  We interpreted this to mean that the editors felt these ideas need further exposure and explication for the chaperone community and we prepared our paper accordingly.

It was our own experience during the review of the NSMB paper, in reading subsequent reviews of Hsp70 disaggregation and force generation mechanisms and in generally discussing these models with biomedical researchers, that these ideas remain poorly understood among biomedical scientists in general, or among the chaperone community in particular.

In support of this we would note that we cited Dr. Eric Kramer, who was published a number of reviews in an effort to correct misconceptions regarding the mechanisms of osmotic force, a topic that is directly related to the misconceptions that make it difficult for scientists to understand entropic pulling.  The need for Dr. Kramer's papers is a clear demonstration of the decades long persistence of these damaging misunderstandings.

In responding to the other reviewers' comments we were able to bring up another example--Peter Mitchell's chemiosmotic hypothesis for the force mechanism underlying ATP synthesis--which was also greeted with skepticism and lack of understanding that similarly reflected confusion regarding biophysics and a strong tendency among biologists and biochemists to seek an 'anthropomorphic' explanation in terms of active species or subjects (high energy chemical intermediates; complex disaggregation engines) rather than in the seemingly vague and weak effect due to the entropy increase that accompanies dissipation of a local concentration gradient.

We could belabor this point further, but we hope our comments above are sufficient and that the editors generally agree with it comes to such subjective editorial decisions as scope and significance.

Since we could not glean any further changes to make to the paper based on this reviewers' current comments, and since the reviewer now seems to be more satisfied with the text of the paper itself, we are not further revising the ms itself.

Sincerely, Rui Sousa & Eileen M. Lafer